# Educational Strategies to Reduce Physician Shortages in Underserved Areas: A Systematic Review

**DOI:** 10.3390/ijerph20115983

**Published:** 2023-05-29

**Authors:** Alexandre Medeiros de Figueiredo, Antonio Olry de Labry Lima, Daniela Cristina Moreira Marculino de Figueiredo, Alexandre José de Melo Neto, Erika Maria Sampaio Rocha, George Dantas de Azevedo

**Affiliations:** 1Department of Health Promotion, Federal University of Paraíba, Campus I, Jardim Universitário, S/N, Castelo Branco, João Pessoa 58051-900, Paraiba, Brazil; 2Health Sciences Postgraduate Program, Federal University do Rio Grande do Norte, Campus Universitário Lagoa Nova, Natal 59078-900, Rio Grande do Norte, Brazil; 3Andalusian School of Public Health, Cuesta del Observatorio 4, Campus Universitario de Cartuja, 18011 Granada, Andalusia, Spain; 4Department of Statistics, Federal University of Paraíba, Campus I, Jardim Universitário, S/N, Castelo Branco, João Pessoa 58051-900, Paraíba, Brazil; 5Health Science Training Center, Federal University of Espírito Santo, Av. Fernando Ferrari, 514, Goiabeiras, Vitória 29075-910, Espirito Santo, Brazil; 6Multicampi School of Medical Sciences, Federal University of Rio Grande do Norte, Av. Cel Martiniano, 541, Caico 59300-000, Rio Grande do Norte, Brazil

**Keywords:** education, medical, medically underserved area, career choice

## Abstract

The shortage of physicians in rural and underserved areas is an obstacle to the implementation of Universal Health Coverage (UHC). We carried out a systematic review to analyze the effectiveness of initiatives in medical education aimed to increase the supply of physicians in rural or underserved areas. We searched for studies published between 1999 and 2019 in six databases, following the Preferred Reporting Items for Systematic Reviews and Meta-Analyses (PRISMA) guidelines. Interventional or observational controlled studies were defined as inclusion criteria. A total of 955 relevant unique records were selected for inclusion, which resulted in the identification of 17 articles for analysis. The admission of students from rural areas associated with a rural curriculum represented 52.95% of the interventions. Medical practice after graduation in rural or underserved areas was the most evaluated outcome, representing 12 publications (70.59%). Participants of these educational initiatives were more likely to work in rural or underserved areas or to choose family medicine, with significant differences between the groups in 82.35% of the studies. Educational strategies in undergraduate and medical residencies are effective. However, it is necessary to expand these interventions to ensure the supply of physicians in rural or urban underserved areas.

## 1. Introduction

The shortage of physicians is considered an obstacle to the implementation of Universal Health Coverage (UHC) especially in less developed regions and rural areas, even in developed countries [1,2,3,4,5]. The distribution of the medical workforce is a complex phenomenon influenced by social, cultural, and economic factors, and therefore, addressing the shortage of physicians in rural and remote areas demands comprehensive policies [3]. 

In 2021, the World Health Organization (WHO) updated their recommendations with strategies aimed at addressing the shortage of health professionals in rural and remote areas [6]. The WHO defined four categories of interventions used to improve the attraction, recruitment, and retention of health workers in remote and rural areas: education, regulations, incentives, and professional and personal support. WHO’s main recommendations concerning education interventions were specific admission policies for students from areas with professional shortages, implementation of medical schools and medical residency programs outside of large cities, learning experiences in rural and underserved areas, curricula that prioritize the development of specific skills to work in rural territories, and access to continuing education programs [6].

Despite the importance of addressing the shortage of physicians to ensure universal health coverage, there are gaps in the evaluation of the effectiveness of interventions carried out in medical education, generating difficulties in guiding the formulation and implementation of public policies. The 2015 Cochrane systematic review identified a limited number of articles [7]. The inclusion criteria established were based on controlled and randomized study designs, not common in medical education studies [7]. A more recent systematic review, restricted to the United States of America, has indicated that graduates of medical schools with experience in clinical training in areas of scarcity are more likely to act as physicians in areas with these characteristics [8].

In addition to the complexity of the factors involved in the distribution of physicians, some authors pointed out the heterogeneity in the definitions of rural and/or remote areas and the structural differences in the organization of Primary Health Care and its related specialties, family medicine (FM) and rural medicine (RM), generating methodological difficulties [9,10].

Several initiatives in medical education implemented by medical schools and governments to reduce the shortage of doctors in rural and underserved areas continue to be implemented and need to be evaluated [5]. The objective of this systematic review was to analyze the effectiveness of strategies in medical education that aim to increase the number of family physicians or physicians practicing in rural or underserved areas in low, middle, or high-income countries. 

## 2. Materials and Methods

This review was conducted in accordance with the PRISMA guideline [11]. We conducted this systematic review of systematic reviews according to a protocol registered in the International Prospective Register of Systematic Reviews (PROSPERO CRD42020172468). The protocol was not published in any peer-reviewed journal.

### 2.1. Eligibility Criteria

Inclusion criteria followed a Population, Intervention, Comparator, and Outcome framework approach. We included studies that analyzed all types of the WHO’s main recommendations concerning educational interventions aimed at addressing the shortage of health professionals in rural and underserved areas.

The searches included articles published between 1 January 1999 and 31 December 2019, in any language and conducted in any country. Observational and interventions studies with control groups were considerate for inclusion. Studies with eligible results were those with any quantitative measure of changes in the number of physicians in rural or underserved areas. Studies without a description of the educational strategy used or that did not describe changes in or the supply of physicians or family medicine physicians were excluded. Unpublished manuscripts, editorials and conference abstracts were not eligible for inclusion.

### 2.2. Data Sources and Search Strategies

Research strategies were created for the PubMed, Web of Science (WoS), EMBASE, ERIC, Cochrane, and LILACS databases. The strategy combined MeSH (Medical Subject Headings) terms and keywords, being initially developed for PubMed (Appendix B). The search strategy was carried out in other databases with the support of a specialist librarian. The database searches were conducted in March 2020. Additionally, cross-references in the study reports included in this systematic review were performed to identify other relevant articles not captured by the initial search.

### 2.3. Selection Process

All articles found were imported to a reference manager by one of the researchers (Rayyan QCRI) and duplicated articles were excluded [12]. These references were distributed for evaluation by reviewers divided into two groups. Group one was composed of AMF and DCMMF and group two of AMF and AJMN. Each reviewer performed independent evaluations of the titles and abstracts for an initial selection. The articles selected in this phase were analyzed in their entirety to assess whether they met the inclusion criteria. The discrepancies between the group one evaluations were resolved by AJMN and disagreements in group two by DCMM. During the full-text screening, AMF searched the citations of reviews to identify additional articles.

### 2.4. Data Extraction

Two reviewers (AMF and DCMMF) independently extracted the data and summarized the most relevant information of all articles included in the review. This data synthesis included the elements of article identification, description, and strategy type (according to the WHO classification) for increasing physicians in underserved areas, objective of the study, methodology used, phase (graduation or medical residency), main results and the MERSQI score. The results were presented as a group of strategies used to increase the number of physicians in underserved areas. The data of effect measures were the prevalence rates, Odds Ratio (OR) or Relative Risks (RR).

### 2.5. Risk of Bias and Quality Assessment

The studies were evaluated using the Medical Education Research Quality Instrument (MERSQI) [13]. This instrument was developed to assess the methodological quality of research in medical education, being used as a reference in several studies [14,15,16].

The instrument presents six domains that evaluate the study design, sampling quality, data collection, characteristics regarding the validation of the research instrument used, quality of the data analysis, and the analyzed outcomes [13]. The maximum score for each domain is three points, with the final score varying from 5 to 18 points [13]. According to previous studies, publications with scores ≥ to 14 were considered good quality studies [16]. Two reviewers (AMF and DCMMF) independently evaluated the selected articles using the MERSQI instrument. The discrepancies between these evaluations were resolved through the mediation of a third evaluator (AJMN). The information regarding the analysis of the quality of the articles was aggregated to the narrative synthesis.

### 2.6. Data Syntesis

A narrative synthesis was conducted. The results were presented as a group of recommended educational strategies to increase the number of physicians in rural and underserved areas.

## 3. Results

### 3.1. Study Selection

The results of the research are summarized in the PRISMA flow chart (Figure 1). A total of 1131 publications were identified in the databases and 15 were identified in the cross-referencing. Screening the title and summary of the remaining 955 publications resulted in the inclusion of 56 citations for later review. After reviewing the full text articles, we excluded 39 articles for a variety of reasons and included 17 articles (Figure 1).

### 3.2. Study Characteristics

The characteristics of the selected publications are described in Table 1. Most of the studies were published during or after 2009 (70.59%). Cohort studies accounted for 76.47% of the publications and cross-sectional studies for 23.53%. Most studies (82.36%) presented an analyses of the interventions that occurred at graduation (Table 1). A summary of the characteristics of the interventions, objectives, methodologies, and results are described in Appendix A.

The 17 studies included in the review represent the analyses of 13 distinct experiences located in six countries (United States, New Zealand, Japan, People’s Republic of Congo, Australia, and Thailand). Almost 65% of the publications refer to experiences developed in the United States. Most publications (82.35%) describe interventions conducted in Rural Programs (RP) [17,18,19,20] developed by educational institutions with the objective of increasing the number of physicians in rural areas or underserved areas. The follow-up time for graduates of these experiences ranged from 2 to 37 years.

In only two studies were instruments used to collect information about the physician’s practice location or the choice of medical specialty. The other 15 publications (88.24%) extracted this information from secondary data.

Admission of students from rural areas associated with curricular activities with emphasis on rural practice, represented 52.95% of the publications analyzed. The practice in rural or underserved areas was the most evaluated outcome, representing 12 publications (70.59%). The definition of what is a rural or underserved area was heterogeneous, being based mostly on criteria established by government agencies or population criteria.

### 3.3. Risk of Bias and Quality Assessment

The result of the quality evaluation using the MERSQI instrument is shown in Appendix A. The MERSQI score was ≥ at 14 points in 47.06% of the studies. The studies that analyzed the association of the admission of students from rural areas with practice-oriented curriculum activities in rural areas presented the best score on the MERSQI scale (14.7). The worst scores were found in publications that evaluated the insertion in rural areas of family medicine residency graduates (12.7).

### 3.4. Study Findings

The studies reported four broad categories of medical education interventions: specific admission programs for students from areas with professional shortages, implementation of medical schools and medical residency programs outside of large cities, and learning experiences in rural and underserved areas.

#### 3.4.1. Specific Admission Programs for Students from Rural or Underserved Areas

Rabinowitz published three studies evaluating the Physician Shortage Area Program (PSAP) and demonstrated that those enrolled in the program were more likely to remain in rural areas after 11 to 16 years (*p* = 0.03) and at 20–25 years (PSAP 70.3% vs. No-PSAP 46.2%, *p* = 0.02) [17,18]. The third study showed that PSAP graduates represented 12% of all Family Physicians in rural Pennsylvania [19]. PSAP graduates were more likely than their non-PSAP colleagues to practice in a rural area (34% vs. 11%; RR, 3.0), underserved area (30% vs. 9%; RR, 3.2), to practice FM (52% vs. 13%; RR, 4.0), and to practice FM in a rural area (21% vs. 2%; RR, 8.5) [19].

MacDowell and collaborators demonstrated that graduates from the Illinois Rural Medical Education (RMED) Program were more likely to choose FM (OR: 14.38; 95% CI: 10.16–20.35; *p* < 0.001) and more likely to be practicing in a rural setting (OR: 17.20; 95% CI: 12.18–24.35; *p* < 0.001) when compared to graduates from the same university not participating in RMED [20]. On the other hand, a study showed that 47% of the graduates of the Rural Program at the Duluth campus specialized in FM, while the average in the United States was about 10% [21].

Another study developed by Rabinowitz and collaborators indicated that 63.8% of the graduates of three Rural Programs in American medical schools (located in Pennsylvania, Minnesota, and Illinois) were working in rural areas, while the percentage of doctors trained abroad was 26.5% [22]. Students who participated in the Rural Programs presented a greater chance of performing FM in rural areas than doctors who were trained abroad (RR: 10.0; 95% CI: 8.7–11.6; *p* < 0.001) [22].

In New Zealand, Shelker and collaborators demonstrated that graduates who were admitted because they came from rural areas or participated in practices in rural areas were more likely to choose postgraduate degrees in a rural hospital or FM (OR: 2.1; 95% CI: 1.2–3.6; *p* = 0.01) [23]. Matsumoto and collaborators showed that 8.7% of the graduates of the Jichi Medical University were practicing in rural areas, after the end of mandatory contracts to work in rural areas, while only 1.1% of Japanese doctors worked in this type of territory [24]. Techakehakij and collaborators demonstrated that graduates from a similar program in Thailand were more likely to stay in hospitals in rural areas (OR: 2.44; 95% CI: 2.19–2.72) when compared to students not participating in this program [25].

#### 3.4.2. Internships in the Clinical Cycle in a Rural Area or Insertion in a Regional Campus

Crump and collaborators demonstrated that physicians who completed one year of their undergraduate clinical training in a small town had a greater chance of choosing FM as a specialty (OR: 4.50; 95% CI: 3.06–6.63; *p* < 0.001) or practice in rural areas (OR: 6.27; 95% CI: 4.26–9.24; *p* < 0.001) when compared with doctors trained in the central campus [26]. A study conducted in Australia presented similar results, pointing out that physicians who undertook their internship in rural areas presented six times more chance (OR: 6.1; CI 95%: 2.9–12.6; *p* < 0.001) of practicing in rural areas when compared to students who stayed in an urban environment [27]. Another Australian study, however, did not find significant differences in the choice of FM as a specialty when it compared graduates with insertion in rural areas and the group of graduates that remained in urban areas [28].

Halaas and collaborators describe the experience of the University of Minnesota (UMN) Rural Physicians Associate Program (RPAP) [29]. The percentage of students who chose FM among the RPAP graduates was almost 70% in 1996 and decreased to 60% in 2006 [29]. Despite this drop, the percentage of graduates from RPAP was higher than the percentage of graduates from UMN that did not participate in RPAP and chose FM during the entire period (19.9% in 1996; 7.9% in 2006) [29]. One of the studies showed that students who completed the first two years of medical school on regional campuses were more likely to practice medicine in primary care and underserved areas compared to students who completed their entire course on a campus located in a large city [30].

#### 3.4.3. Medical School in a Rural Area

One study showed that 81.4% of graduates from a medical school located in an underserved area were practicing in rural areas at the time of the study’s analysis, while the percentage of graduates from a medical school located in an urban area was 26.7% (*p* < 0.01) [31].

#### 3.4.4. Medical Residence with Rural Training Sites

A study by Fergunson and collaborators pointed out that graduates of family medicine that were residence trained in rural areas presented a greater chance of starting their careers practicing in rural areas (OR:5.61; CI95%: 2.01–15.7) or underserved areas (OR:4.53; CI95%: 1.43–14.35) when compared to residents who had completed their training in urban hospitals [32]. The results of the publication by Pacheco and collaborators showed that 65.1% of graduates from residencies in family medicine located in rural areas began to practice medicine in rural areas after completing their training, while the percentage of residents who completed their training in the urban zone it was 25.8%(*p* < 0.001) [33].

### 3.5. Results of Synthesis

The evaluated studies demonstrate that the four types of educational intervention recommendations analyzed are associated with an increase in the number of physicians in rural or underserved areas. The methodological heterogeneity did not allow the realization of a meta-analysis.

## 4. Discussion

Our systematic review identified 17 publications on medical education interventions that aimed to increase the number of physicians in rural or underserved areas. It is observed that recent studies have obtained better scores, indicating improvement in the quality of studies over time.

These results indicate that specific admission programs of students from rural areas have been effective in increasing the number of physicians in rural areas [17,19,20,22,25,26,27] and the number of FM specialists [17,19,20,22,23,25,28]. The greater chance of practicing in rural or underserved areas seen in the participants of specific programs for admission to medical schools can be explained by the influence of personal characteristics on decision making. Studies indicate that other personal characteristics, such as the physician’s race and ethnicity, are among the strongest predictors for choosing a specialty and for choosing to practice in areas with vulnerable populations [34].

Only one selected publication evaluated the influence of the implantation of a medical school in an underserved area, showing the effectiveness of this measure [31]. The implantation of a medical school in rural or underserved areas can increase the access of the populations that live in these areas to medical education. The expansion of medical schools to rural or underserved areas can also expand health services, improving working conditions and increasing the capacity of these areas to retain physicians [3]. Interventions during medical residency have also proven to be effective in increasing the number of physicians in rural and remote areas [29,30]. Thus, it is possible to observe that interventions at various moments during medical training can be used in isolation or integrated.

Most of these interventions have a regional scope (88.24%) and the number of annual graduates from these programs represents only a small contingent, when compared to the total number of graduates from medical courses. Thus, despite the local impact on the increase in the medical workforce, these initiatives have not generated enough graduates to reverse the iniquities of the regional distribution of physicians and situations of scarcity in rural areas or remote areas, even in countries where they have been implemented [35,36,37,38].

Selected publications describe interventions in a small group of countries, mostly in the US and other developed countries. This greater number of publications in developed countries may be related to the greater organization of civil society and the demand for reducing inequalities in access to health in underserved areas. In the case of the USA and Australia, the vast rural and/or remote areas increase the need to develop physician retention strategies, demanding greater attention from governments, medical schools, and researchers. Another aspect worth mentioning is the greater number of opportunities for disseminating existing experiences in developed countries. The limited number of publications on experiences in developing countries reproduces the challenges of ensuring medical training in poorer countries and the difficulty of conducting research and getting it published [39,40]. This finding is important, because, according to the last UN report, almost 40% of the countries in the world have fewer than one physician per thousand inhabitants, especially in this group of countries [2].

Despite presenting a good MERSQI score, all evaluated studies present some degree of selection bias because the admission processes of Rural Programs seek students from rural areas, or because the adherence to immersion experiences in rural areas was optional. Thus, there are limitations in defining whether the choice to practice in rural or underserved areas or to choose family medicine was caused by the educational strategy or whether personal characteristics and socioeconomic factors influenced these choices. Another methodological aspect is that the studies did not consider the use of variables related to the labor market to identify other elements of influence in the decision to practice in rural or underserved areas. Understanding the relationship between educational policies and the organization of health systems is important due to the interdependence between these two sectors and the well-known relationship between economic development and the regional supply of doctors [39,40]. This review only included studies on the increase in physicians or family physicians in underserved areas and, therefore, did not evaluate strategies concerning the expansion of physicians of other specialties in underserved areas. We suggest further studies to identify strategies in medical education to increase the number of other specialists in underserved areas.

Ethnic minorities and populations with low income also have difficulties in accessing healthcare, even when they live in urban areas [39]. Despite the possibility of inclusion, we did not identify studies on educational strategies to improve physician retention in underserved urban areas. Finally, it is possible that important experiences have not been published or identified by the search strategy used.

## 5. Conclusions

Our results demonstrate the effectiveness of medical education interventions in increasing the number of physicians in rural or underserved areas and increasing the number of doctors who choose FM as a specialty. However, reversing the deficit of physicians in underserved areas requires an expansion in the number of institutions that use these types of interventions and an increase in the number of participants. It is important to emphasize that economic aspects are relevant in the expansion of these initiatives, which require the formulation of public policies, especially in low-income regions.

Follow-up of the experiences described, and additional research to identify other interventions and their impacts on the workforce is recommended, especially in middle- and low-income countries. We suggest that further studies be carried out to identify educational strategies to increase the number of physicians in underserved urban areas and low and middle-income countries. Additionally, the development of studies with comprehensive approaches that aggregate variables related to the labor market to the variables commonly used in the field of medical education are sugggested.

## Figures and Tables

**Figure 1 ijerph-20-05983-f001:**
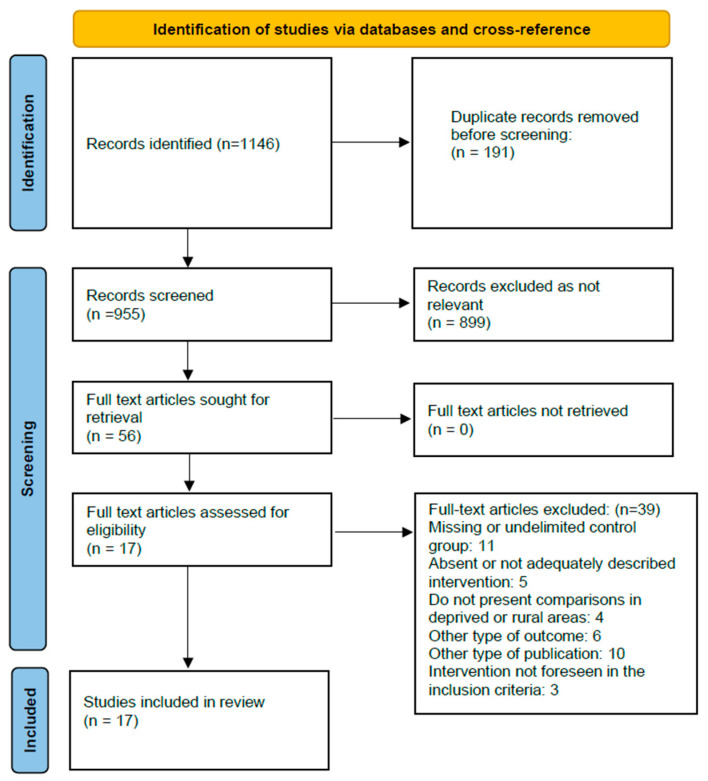
Flowchart of the study selection process.

**Table 1 ijerph-20-05983-t001:** Characteristics of the studies selected in the systematic review.

Characteristics of the Studies Evaluated	Number (%)	Average Score MERSQI
Year of publication		
1999–2008	05 (29.4)	13.3
2009–2019	12 (70.6)	14.3
Study design		
Cohort study	13 (76.5)	14.5
Cross-sectional study	04 (23.5)	12.2
Medical training		
Graduation	14 (82.4)	14.2
Medical residency	02 (11.8)	12.7
Graduation and medical residency	01 (5.9)	13.2
Type of educational experience		
Admission of students from rural areas associated with curricular activities with emphasis on rural practice	09 (52.9)	14.7
Internships in the clinical cycle in a rural area or insertion in a regional campus	05 (29.4)	13.3
Medical school in a rural area	01 (5.9)	13.8
Medical residence with rural training sites	02 (11.8)	12.7
Evaluated outcome		
Practice in rural or shortage areas	06 (35.3)	13.7
Specialization in Family and Community Medicine	05 (29.4)	12.6
Both	06 (35.3)	15.4
Total	17 (100.0)	14.0

## Data Availability

Not applicable.

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
