# Peer review of "Educational Strategies to Reduce Physician Shortages in Underserved Areas: A Systematic Review"

_ijerph, 2023, doi:10.3390/ijerph20115983_

Round 1
Reviewer 1 Report
Excellent study. Minor point - when using numerals, do not use leading zeros (use 6, not 06). It would be worth mentioning the "natural experiment" of NYU Medical School's tuition-free adoption thanks to a huge grant - see if specialty choice changes significantly from historical pattern.
Author Response
Dear reviewer,
First, we would like to thank you for your comments and revision suggestions that made it possible to improve the text and quality of the article. We made changes throughout the text, mainly in the methodology and results sections, aiming to improve the description of the methods used and results found. There was an adaptation to the latest version of the Prism Check List. The discussion section was expanded with the deepening of issues already described in the first version and new reflections made from the reviewers' comments. In response to your suggestions:
Point 1 - Minor point - when using numerals, do not use leading zeros (use 6, not 06).
Response 1: We excluded leading zeros.
Point 2 - It would be worth mentioning the "natural experiment" of NYU Medical School's tuition-free adoption thanks to a huge grant - see if specialty choice changes significantly from historical pattern.
Response 2: Thank you for featuring NYU Medical School's free adoption initiative. These and other initiatives could form part of future analyses.
Best Regard

Reviewer 2 Report
Thank you very much for the manuscript. In the document, the authors address the available evidence on different strategies in relation to medical education implemented to increase the availability of doctors in rural areas.
The search for information is clearly stated and articles that provide valuable information on the subject were identified, also acknowledging the limitations of the review.
Here are some minor recommendations to include in the document:
title: it is suggested to include in the title "in medical education". In this way, false expectations are not created, and it is more consistent with the objective.
Materials and Methods: Line 68, include PROSPERO registration number
Line 98 typo _ "... as good quality studies16."
Discussion: there will be some limitations in relation to the outcome considered (eg, in specialties only family doctors were considered).
Author Response
Dear reviewer,
First, we would like to thank you for your comments and revision suggestions that made it possible to improve the text and quality of the article. We made changes throughout the text, mainly in the methodology and results sections, aiming to improve the description of the methods used and results found. There was an adaptation to the latest version of the Prism Checklist. The discussion section was expanded with the deepening of issues already described in the first version and new reflections made from the reviewers' comments. In response to your suggestions:
Point 1: Title: it is suggested to include in the title "in medical education". In this way, false expectations are not created, and it is more consistent with the objective.
Response 1: We agreed with the suggestion and changed the title to: “Educational strategies to reduce physicians’ shortages in underserved areas: a systematic review”.
Point 2: Materials and Methods: Line 68, include PROSPERO registration number.
Response 2: Thanks for the comment, the registration number on Prospero was included in the main text.
Point 3: Line 98 typo _ "... as good quality studies16."
Response 3: Thanks for the comment, the text has been rewritten for better understanding.
“According to previous studies, publications with scores ≥ to 14 were considered good quality studies [16]”
Point 4: Discussion: there will be some limitations in relation to the outcome considered (eg, in specialties only family doctors were considered).
Response 4: Thanks for the comment, we included this limitation in the discussion section.
“This review only included studies on the increase of physicians or family physicians in underserved areas and, therefore, did not evaluate strategies on the expansion of physicians of other specialties in underserved areas. We suggest studies to identify strategies in medical education to increase the number of other specialists in underserved areas.”
Best Regard

Reviewer 3 Report
The manuscript entitled “Strategies to increase the supply of physicians in underserved areas: a systematic review” was interesting. However, there are some major concerns:
1- Please choose appropriate keywords using MeSH terms.
2- When you are using more than one reference and they are in sequence, please show them like [1-5].
3- Please use the headings and subheadings suggested in the PRISMA checklist to report the methods and results sections.
4- We usually use the same search strategies in different databases. However, the search strategies used in the current study are different.
5- “Text words” and “keywords” may increase the number of the retrieved papers which are not necessarily relevant to your research, and they can be used just once in a manuscript.
6- Please use the latest version of the PRISMA flow chart for systematic reviews (2020).
7- At the end of the results section, please add a “synthesis section”.
8- In Table S1, please report the studies in a chronological order.
Author Response
Dear reviewer,
First, we would like to thank you for your comments and revision suggestions that made it possible to improve the text and quality of the article. We made changes throughout the text, mainly in the methodology and results sections, aiming to improve the description of the methods used and results found. There was an adaptation to the latest version of the Prism Checklist. The discussion section was expanded with the deepening of issues already described in the first version and new reflections made from the reviewers' comments. In response to your suggestions:
Point 1: Please choose appropriate keywords using MeSH terms.
Response 1: We agreed with the suggestion and changed the keywords using MESH terms.
Point 2: When you are using more than one reference and they are in sequence, please show them like [1-5].
Response 2: Thanks for the comment, we've made changes throughout the text.
Point 3: Please use the headings and subheadings suggested in the PRISMA checklist to report the methods and results sections.
Response 3: We agree with the suggestion. We have changed headings and subheadings in line with the PRISMA checklist for reporting methods and results sections.
Point 4: We usually use the same search strategies in different databases. However, the search strategies used in the current study are different.
Response 4: We carry out different search strategies for each database to qualify the searches, since the search process is sometimes different in these databases. Adaptations were guided by a specialized librarian. We detail this choice in more detail in the methodology section.
Point 5: “Text words” and “keywords” may increase the number of the retrieved papers which are not necessarily relevant to your research, and they can be used just once in a manuscript.
Response 5: We chose to perform a more comprehensive search in the databases to reduce the risk of losing important articles.
Point 6: Please use the latest version of the PRISMA flow chart for systematic reviews (2020).
Response 6: We agree with the suggestion. We have updated the figure according to latest version of the PRISMA flow chart for systematic reviews.
Point 7: At the end of the results section, please add a “synthesis section”.
Response 7: We agree with the suggestion and included a synthesis section.
“3.5. Results of synthesis
The evaluated studies demonstrate that the four types of educational intervention recommendations analyzed are associated with an increase in the number of physicians in rural or underserved areas. The methodological heterogeneity did not allow the realization of a meta-analysis.”
Point 8: In Table S1, please report the studies in a chronological order.
Response 8: We agree with the suggestion and reported the studies in a chronological order.
Best Regard

Round 2
Reviewer 3 Report
I appreciate the authors for their time and efforts to revise the manuscript. However, some issues are still remaining.
1-In PRISMA flow chart, you can report the total numbers for each box and it is not necessary to separate the chart for the studies identified via other methods.
2- I was not convinced by the authors regarding the use of different search strings for different databases. In my opinion, the search string should be the same in different databases and the advance search feature of each database needs to be used along with selecting appropriate filters to reach the valid results.
Author Response
Dear reviewer
First, I would like to thank the revision suggestions, as they would generate significant and productive changes to the article. In response to your recomendations:
Point 1: -In the PRISMA flowchart, you can report the total numbers for each box and it is not necessary to separate the graph for the studies identified by other methods.
Answer 1:
We've changed Prisma's flowchart as recommended.
Point 2 - I was not convinced by the authors regarding the use of different search strings for different databases. In my opinion, the search string must be the same in different databases and the advanced search feature of each database needs to be used along with selecting appropriate filters to reach the valid results.
Answer 2:
We use the same search terms across all databases. Boolean operators were used in the same way in all databases. When we refer to different search strategies, we are referring to the filter selection process in advanced searches in these different databases. This process was supported by an experienced librarian. We changed the text in the methodology section as below:
"The search strategy was carried out in other databases with the support of a specialist librarian"
Best Regard